# Revealing the mechanism and function underlying pairwise temporal coupling in collective motion

Guy Amichay [1,2,3,4,5] ✉, Liang Li [1,2,3], Máté Nagy [1,2,3,6,7] ✉ &
Iain D. Couzin [1,2,3] ✉

Coordinated motion in animal groups has predominantly been studied with a focus on spatial interactions, such as how individuals position and orient themselves relative to one another. Temporal aspects have, by contrast, received much less attention. Here, by studying pairwise interactions in juvenile zebrafish (*Danio rerio*)—including using immersive volumetric virtual reality (VR) with which we can directly test models of social interactions in situ—we reveal that there exists a rhythmic out-of-phase (i.e., an alternating) temporal coordination dynamic. We find that reciprocal (bi-directional) feedback is both necessary and sufficient to explain this emergent coupling. Beyond a mechanistic understanding, we find, both from VR experiments and analysis of freely swimming pairs, that temporal coordination considerably improves spatial responsiveness, such as to changes in the direction of motion of a partner. Our findings highlight the synergistic role of spatial and temporal coupling in facilitating effective communication between individuals on the move.

Collective motion underlies many important biological processes across scales in biology[1]. As a consequence, the mechanisms that give rise to coordinated motion have received considerable attention, connecting theoretical principles[2–6] to experimental data for a wide range of systems, including cells forming tissues[7,8], the onset and maintenance of swarming in insects[9–11], schooling in fish[12–19], flocking in birds[20–22] and the formation of mobile aggregations in humans[23–25].

Such studies have emphasized the role of relatively local (and thus self-organized) spatial interactions among system components (e.g. cells/organisms), with evidence having been found for multiple types of spatial interaction, including those that depend on distance (so-called metric interactions)[26], a fixed number of individuals irrespective of their distance (topological interactions)[20] and

interactions that explicitly depend on sensing, such as visual perception, of others (sensory interactions)[15]. Consequently, it is widely appreciated that the spatial structure of interactions has important consequences to information flow in groups, such as waves of changing velocity that characterise response to both attractive and aversive stimuli, such as stimuli associated with food[15] or a localised threat[27], respectively.

By contrast, relatively little work has considered the role of temporal coupling (coupling in time) in the regulation of collective motion, or the feedback processes that may exist between spatial and temporal coupling in groups. In schooling fish, a common model for investigating collective motion among animals[28], it has, however, been shown that some species do regulate temporal aspects of their

[1]Centre for the Advanced Study of Collective Behaviour, University of Konstanz, Universitätsstraße 10, 78464 Konstanz, Germany. [2]Department of Collective Behaviour, Max-Planck Institute of Animal Behavior, Konstanz, Germany. [3]Department of Biology, University of Konstanz, Konstanz, Germany. [4]Department of Engineering Sciences and Applied Mathematics, Northwestern University, Evanston, IL, USA. [5]Northwestern Institute on Complex Systems, Northwestern University, Evanston, IL, USA. [6]MTA-ELTE Lendület Collective Behaviour Research Group, Hungarian Academy of Sciences, Budapest, Hungary. [7]ELTE Eötvös Loránd University, Department of Biological Physics, Budapest, Hungary. ✉e-mail: guy.amichay@northwestern.edu; nagymate@hal.elte.hu; icouzin@ab.mpg.de

interactions. Banded killifish (*Fundulus diaphanous*), for example, exhibit regular, periodic oscillations in swim speed (that occur over a slower timescale than their tail beat frequency), and exhibit a tendency to remain out of phase in relation to their nearest neighbours. It was speculated that this systematic cycling of relative spatial positions among near neighbors may allow fish to better detect changes in the speed and/or heading of others[29]. Evidence has been found that suggests that adult zebrafish (*Danio rerio*), while not exhibiting clear oscillatory dynamics, exhibit burst-and-glide dynamics in which turns are made during the active acceleration, and inertia and friction are largely responsible for the deceleration (the passive glide). It is argued that when fish are passive during decelerations, they may collect valuable social information, but that it is during bursts when this information is enacted upon. Thus the response to others may occur in limited time periods, but the timescale where information is obtained from the environment could not be explicitly determined[17].

For juvenile fish, the relatively high viscosity of the water results in burst-and-glide becoming extremely pronounced-appearing as highly abrupt jerky locomotion. As with other sensory systems that must exhibit sudden movements, such as during eye saccades, environmental sensing has been found to be severely compromised during periods of high acceleration and speed[30]. As a consequence, sensory suppression, and/or cancellation of neural outputs during such periods (reafferent cancellation) is ubiquitous across sensory modalities and species[31,32]. For sensing via their lateral line (which detects flow and pressure changes in water in very close proximity to the body) it has recently been shown that young (7 days post fertilization) zebrafish already exhibit a graded subtraction of self-generated motion, thus maximally-silencing reafferent sensory signals during the most vigorous tail activity during each burst[33]. Rapid self-generated motion will also tend to induce motion blur to the visual field that are likely not possible to compensate for by eye movements (which also introduce motion blur−silenced through saccadic masking)[34].

Thus, despite multiple possible mechanisms having been suggested, the nature and functional consequences of time-varying sensing and motor response in regulating collective behavior remain largely unknown. One of the key issues has been that establishing the time-varying reciprocal coupling of interaction strength among individuals, even when only considering a pair, proves very challenging. For example, motor decisions being made in relatively discrete windows of time, does not inform us about possible windows of perception, or the timescale that informs each discrete motor decision, is obtained[17]. The changing strength and direction of reciprocal social coupling can make the causal time-varying structure of interactions hard to infer.

Recent advances in virtual reality (VR) technology for freely-moving animals, however, offer the possibility to both control the causal structure of social relationships among individuals, including insects, mammals, and fish[18,35], and to test specific hypotheses regarding the nature of social interactions in-situ. Thus, similar to how the dynamic patch clamp method has allowed principled exploration of the reciprocal coupling between neurons, virtual reality sets the scene for a dynamic social clamp paradigm. First proposed to study real-time human interactions, but emphasized to be a powerful tool across systems and scales of biological organization[36], this approach allows real-time bidirectional interactions between animals and empirically-derived, or empirically-grounded, models of coordination dynamics.

Here, we employ this dynamical social clamp approach to reveal key aspects of temporal social coupling during the regulation of collective motion, and in response to sudden changes in salient social stimuli, using pairs of interacting juvenile zebrafish (24−26 days post fertilization; body length of 9-11 mm) as our model biological system. Understanding the coordination of pairs of individuals is a valuable starting point, both for tractability, due to the rich dynamics we see in

pairs, and because swimming in pairs is the most common configuration found in most natural fish populations[28], and it has been found that even when schooling individuals tend to swim close to, and behaviorally couple most strongly with, a single neighbor[37].

By embedding real fish into immersive, volumetric (holographic-like) environments in which they can freely interact with a virtual conspecific (with whom they interact in the same way as they do to real conspecifics[18]), whose motion can be precisely controlled in both open- and closed-loop (see[18,35] for details) we were able to overcome the inherent limitations of purely observational studies. Combining this with traditional experiments, we can (i) establish, from data of freely swimming pairs, an experimentally-derived model of how zebrafish couple behavior, temporally, with a partner, and (ii) (in both open- and closed-loop) test our experimentally-derived model, revealing the importance of bidirectional temporal coupling in regulating naturalistic collective motion, and, finally, (iii) (in open-loop) demonstrate the functional significance of temporal interactions in facilitating effective response to rapidly-changing spatial information −sudden changes in direction−a vital feature for effective motion coordination in mobile groups (see Supplementary Fig. 1 for an overview of the methodology).

## Results

### Temporal coordination in pairs of juvenile zebrafish

Freely-swimming juvenile zebrafish swim in characteristic pulsatile movements corresponding to their burst-and-glide gait (Fig. 1A). When relatively far (> 6−7 cm) from each other, such-aged zebrafish do not school (with a sharp decrease in social interaction strength at around 5 cm, and a tendency to swim on the same plane; see[18]). As is common among many fish species, individuals exhibit periods of time of highly-coordinated motion when relatively close, and uncoordinated motion when relatively far (termed fission-fusion dynamics[28,38]) while typically swimming with different speeds (Fig. 1B). Here we tracked (using TRex[39]) 29 pairs of zebrafish swimming freely in two different arenas: a square 30 × 30 cm² arena, and a circular arena with a diameter of 28.7 cm, with water depth -0.5 cm in both cases.

Fish were considered to be socially-isolated using a conservative inter-individual distance of > 10 cm[18]; we also confirmed this separation by comparing the cross-correlation between close-by, far-away, and randomly shuffled pairs of fish, and their behavior differed significantly (Fig. 1C; Kolmogorov-Smirnov test: p < 0.001, refer to Fig. S2 for a detailed explanation of the correlation technique and Fig. S3 for the statistical procedure). While each fish exhibits variable speed over time (due to their gait), the time intervals (lags) between successive bursts also exhibit a broad distribution of values (Fig. 1D).

When in close proximity and schooling most strongly−at swim speeds that are above their typical speed when isolated, of > 6 cm/s (conditions under which social interactions in zebrafish, and other species are most pronounced, see:[14,16,40]), fish are found to exhibit prominent temporal coupling with respect to the timing of their bursts, exhibiting a characteristic (shared) time-lag, and out-of-phase (i.e., alternating) relationship, between bursts (as evident in the symmetric curve in Fig. 1C). This temporal coupling is characteristic of out-of-phase coupled oscillators, a common temporal motif underlying the regulation of collective dynamics of other biological systems, including in some conditions neurons[41], among neural groups[42] or between humans[43].

We note that regular temporal dynamics at a collective level need not imply strong, or any, rhythmic behavior on the part of system components (such as juvenile zebrafish, which show irregular, and thus non-rhythmic bursts; Supplementary Fig. 4). Sustained rhythmic coupling between system components can, for example, emerge spontaneously when these components mutually excite one another (such as a neuron's firing increasing the probability of another to whom it is connected to subsequently fire), but also exhibit a

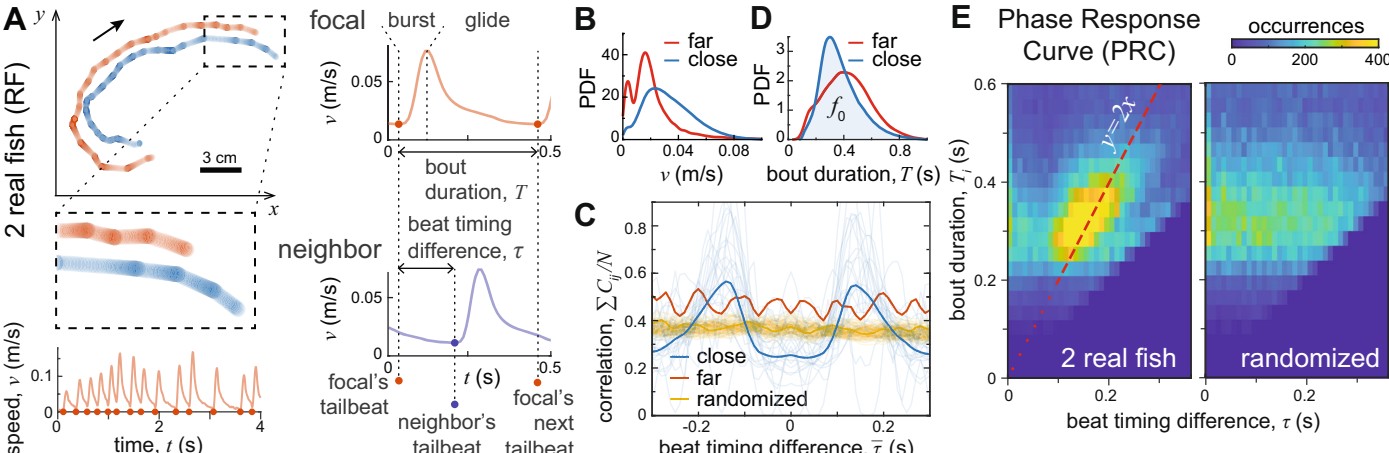

**Fig. 1 | Temporal relationship of real fish (RF) pairs. A** 2D trajectories of two RF swimming together for 10 s and a graphical explanation for obtaining the important temporal parameters by locating the timing of the burst (beat timing). The beat timing signal was obtained from the speed over time. Dots denote the time the fish started a bout, when a local minima occurred in the speed profile. These time points define the bout duration $T$, and for pairs, the beat timing difference $\tau$. **B** Occurrences (probability density function; PDF) of mean swimming speed for individuals when their conspecific was in close proximity ( < 4 cm) or far away ( > 10 cm). **C** The correlation function $C_{ij}$ between 2 RF normalized with the number of sample windows for pairs swimming close (blue), far (red) or for shuffled pairs generated by randomization (yellow; randomizing the entire dataset 100 times,

resulting in 100 different curves; see the methods section for more details). Thick lines show single curves from pooled data, and thin, semi-transparent curves show data separately for each pair. **D** Occurrences (PDF) of bout durations.The distribution $f_0$, which was derived from times when the pairs were close, is the naturally occurring distribution that was used in our model (see later; Fig. 2B). **E** Heatmap showing the relative occurrences of the beat timing difference $\tau$ and bout duration $T$. If a relationship exists and $T$ is a function of $\tau$, then that defines a phase response curve (PRC). $y = 2x$ is shown for reference as a dashed red line overlaid on the heatmap (left). For comparison, we show the resultant heatmap when we create shuffled pairs by randomizing (right).

---

refractory period (such as the inability for a neuron to fire again due to the necessity for ions to be transported back into the cell). In other neural systems it arises when cells inhibit one another but then exhibit post-inhibitory rebound[44], with alternating bursting of neural activity being a dynamically-stable state resulting from the interactions[45]. Similar dynamics are ubiquitous and are found also in animal collectives, such as ants (where isolated individuals exhibit temporally chaotic activity, but as a colony collectively synchronize activity via contact-based mutual excitation)[46–48] or spiders (which exhibit synchronized movements in pursuit of prey)[49–51].

### Quantifying the phase response curve (PRC) suggests a putative mechanism of coupling

To establish insights into the mechanism underlying the observed out-of-phase coupling in our paired-fish system we first quantified its phase response curve (PRC)[52,53]. PRCs have been employed to characterize a wide range of oscillatory systems, including cardiac rhythms[54], networks of neurons[55], coupling of circadian clocks[56], and in temporal coordination of animal signalling, such as flashing in fireflies[57] and singing in crickets[58]. For both linear and nonlinear dynamical oscillating systems valuable information regarding the mechanics of interactions can be extracted by studying the individual components—in our case the individual fish—and simplifying their oscillatory behavior (if appropriate) to single pulses. Note that this does not make any implications about which aspect of the oscillating cue/signal is causally influencing the response, rather it allows us to quantify the overall temporal relationships exhibited, from which we can then employ biological insights to generate testable hypotheses.

Specifically, the PRC of our system can evaluate the transient change (phase response) in a focal individual's bout duration—the time lag ($T$) of a focal fish ($i$) between its previous burst and its subsequent burst, i.e., period length—as a function of the delay in time ($\tau$) between its former burst and that exhibited by the other fish. Thus it evaluates how the burst exhibited by an individual to which it is socially-coupled influence its timing of itself producing another burst.

In doing so (Fig. 1E) we see that fish tend to be oblivious (unresponsive), in terms of the timing of their subsequent burst, if their neighbors burst followed theirs closely in time (when $\tau < 0.1$ s) henceforth referred to as the unresponsive temporal window. This is consistent with the hypothesis that during the rapid acceleration and speed of the burst phase, fish may be transiently unable to obtain salient sensory input[59].

Following this short unresponsive interval, the PRC shows fish employ the time difference between their burst, and that observed to modulate the timing of their subsequent burst (up to time lags that exceed the typical timing interval range exhibited, being approximately 0.23 s). We refer to this as the responsive temporal window. We can approximate the observed trend here as a linear relationship of $y = 2x$ (the red dashed line in Fig. 1E). When comparing the $\chi^2$ values of the fit to the actual data with shuffled pairs, we see a significant statistical difference with a Kolmogorov-Smirnov test: $p < 0.001$ (Supplementary Fig. 5). Also, performing a linear regression analysis for the data between 0.1 s < $\tau$ < 0.4 s for each pair separately, we report a mean regression coefficient of 1.71, with $R^2 = 0.48$, showing that a focal individual's response can be approximated as it waiting for the same interval, the $\tau$ it just experienced, again, before exhibiting its own burst. In other words, this simple functional form can enable an out-of-phase relationship to emerge, even amongst irregular (non-isochronous) oscillators. When one reacts early the other will tend follow suit, or when one is late the other will also be inclined to delay its own response.

### One-way information flow is insufficient to explain the observed synchronized temporal structure of real pairs

The above analysis suggests that fish may employ simple rules of thumb to regulate temporal coordination when schooling—but is this explanation sufficient to explain the observed temporal dynamics? If fish keep track of the time interval between their burst, and that of a conspecific ($\tau$), and then add that time interval to their internal timer employed to decide when to produce their next burst, we may expect to see this same timing regulation to be exhibited in open-loop

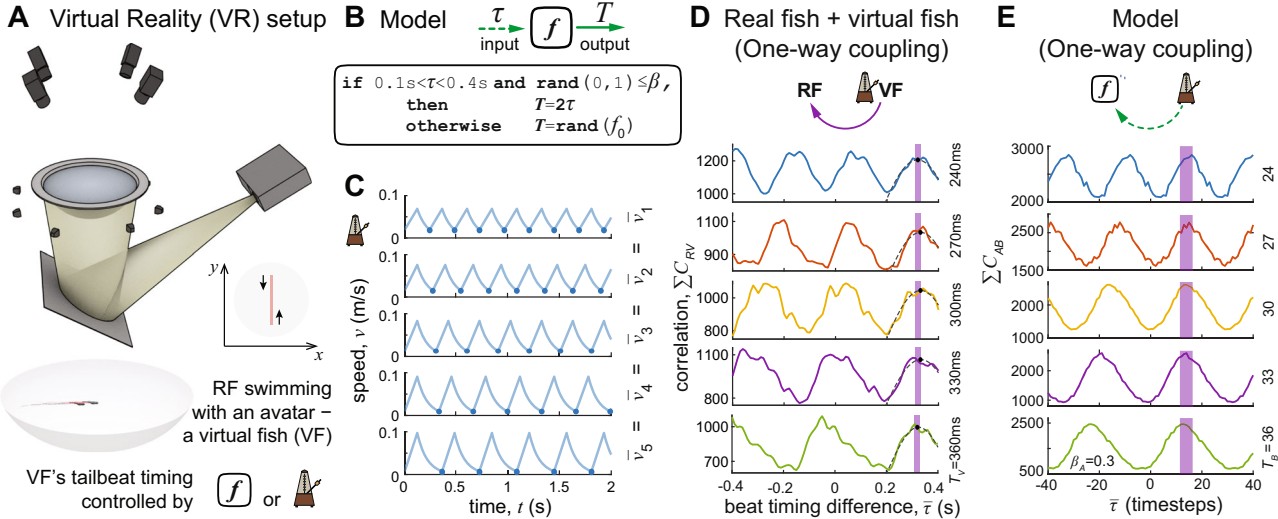

**Fig. 2 | Temporal coupling studied in virtual reality and description of a model to replicate the interactive temporal behavior of real fish. A** Illustration of the immersive virtual reality setup, and the details of the settings used in the experiments. The VF was swimming on a set straight path back and forth, with its burst timing controlled by an interactive (responsive) model or by a set fixed beating pattern (metronome). **B** Illustration of the model, that captures responsive behavior using a PRC of $y = 2x$. Responsive behavior occurs if the beat of the neighbor falls within a time window (between 0.1 s and 0.4 s) and a random stochastic process characterized by $\beta$ (the strength of the PRC term, spanning from 0 to 1: entirely random to no randomness, i.e., the latter situation being fully interactive). Otherwise the bout duration is picked randomly from the naturally observed bout length distribution ($f_0$; Fig. 1D). **C** Speed patterns of VF with fixed inter-burst

intervals (bout durations). This provided a systematic way to study how this timing information influences the RF. The overall travel speed was kept constant, which allows us to vary only one parameter, the duration of bouts. **D** Correlation functions obtained from experiments with a non-interactive VF following a fixed beat timing. Each separate plot corresponds to a different VF bout the duration $T_V$. The peaks on the right were fitted with a Gaussian function denoted by a dashed line, and its peak value with a black dot. Purple bands show the range in which these peaks occurred, which defines the shared lag. **E** Simulations of the experiment (as shown on **D**; data presented the same way) replacing the real fish with our model. One timestep is analogous to 0.01 s in the experiment, but we did not fit this nor aimed to recreate the exact values for the shared lag, rather the qualitative behavior.

experiments with a virtual conspecific (Fig. 2A); i.e., where the real fish can be influenced by the virtual fish, but not vice versa (see closed-loop vs. open-loop in Fig. 2B vs. 2C). In other words, if one-way information flow is sufficient for temporal coupling, we would expect the timing of the focal fish's tailbeats to alternate with those of the virtual fish. Thus, we would expect to see two peaks in the correlation function symmetrical about $\tau = 0$.

To ask whether this is the case we investigated how real fish time their bursts in response to virtual conspecifics which exhibit different inter-burst intervals, across the natural range exhibited by pairs of real fish (240 ms–360 ms, Figs. 1D and 2C). We found that, irrespective of the burst-burst frequency exhibited by the virtual fish, when the real fish interacted with the virtual fish (i.e., were mostly in close proximity with a distance of < 4 cm, as with the analysis of 2 RF) they maintained a constant temporal lag of approximately 0.32 s between the burst of the virtual fish and their own subsequent burst (we fit a Gaussian to each of these peaks to reliably detect their maxima—for each VF period length, with 95% confidence bounds, these were: $T_V$=240 ms: 311.8 ms, ± 5.7 ms; $T_V$=270 ms: 327.6 ms, ±10.8 ms; $T_V$=300 ms: 332.9 ms, ± 8.7 ms; $T_V$=330 ms: 330.9 ms, ± 5 ms; $T_V$=360 ms: 323.3 ms, ± 3 ms) (Fig. 2D). Thus, if only provided unidirectional information flow, zebrafish do not exhibit the out-of-phase temporal coupling observed in real fish pairs (bidirectional information flow; Fig. 1C).

### Computational model of unidirectional and bi-directional temporal coupling

To investigate why this may be the case we developed a simple computational model of the mechanism suggested by the PRC analysis, taking into account the sources of stochasticity observed in the natural system; notably the probabilistic nature of individual response and the irregularity exhibited in the inherent timing of bursts. This model captures the three core features (rules of thumb) suggested by our analyses of real fish pairs (Fig. 1).

Rule 1: Individuals are unresponsive to the burst of another if experiencing it within $t \le 0.1$ s after their own burst ($t = 0$).

Rule 2: If experiencing the burst of another individual after the unresponsive period, but still within a reasonable waiting time ($t < 0.4$ s; to ensure continued motion), individuals have a probability of adjusting the timing of their subsequent burst (if $P \le \beta$, where $\beta$ defines the strength of the interaction), and if they do so they delay it by the estimated timing difference, $\tau$, between these events, i.e., they beat at $t = 2\tau$. Thus their inter-burst interval, $T$ is $2\tau$.

Rule 3: Otherwise, as individuals are inherently stochastic with respect to the timing of their bursts (Fig. 1D); we assume that they employ a stochastic process, based on the observed irregularity of intrinsic inter-burst-intervals exhibited by the fish (Fig. 1D), to determine when to produce their next burst. Thus, Rule 3 captures the intrinsic capability of fish to burst irrespective of social cues.

We first consider the scenario where there exists only unidirectional flow; simulated individual A responds, probabilistically (by following the above rules), to the bursts exhibited by simulated individual B, but not vice versa. That is, B is the driver of the system dynamics, and is assumed to exhibit bursts at a fixed interval, but over a range of frequencies, with high frequencies corresponding to short inter-burst intervals and vice versa (to allow comparison to our open-loop experiments).

Note that we studied the behavior of the model with increments of $\beta$ values over its full range (see Supplementary Figs. 6–9 for multiple parameter space analyses). Since our results are robust to this choice (see Supplementary Figs. 6–9), we present a single value ($\beta = 0.3$) here (e.g., Fig. 2E), for simplicity, and without loss of generality. In addition, our findings are robust to both the starting conditions (initial lag) (Supplementary Figs. 10, 11) and, especially in the presence of stochasticity, to the specific value chosen for the slope of the PRC (Supplementary Figs. 12, 13). This demonstrates that out-of-phase coupling is robust in the face of inherent errors associated with perception and action.

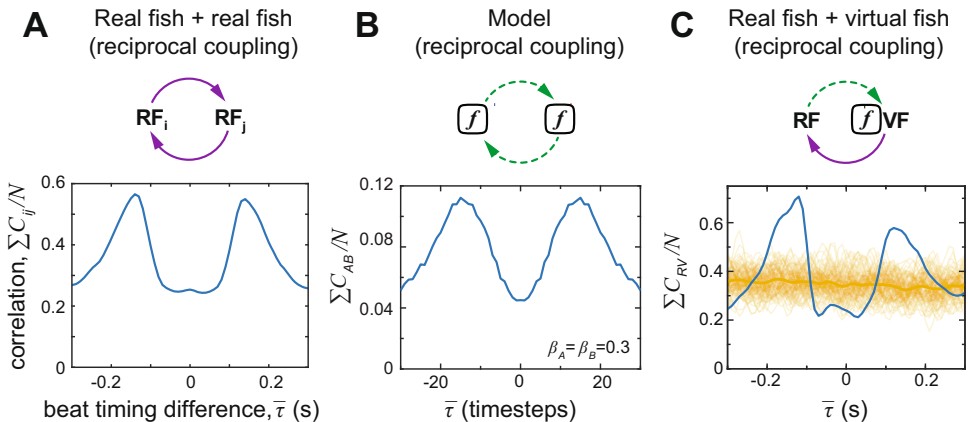

**Fig. 3 | A comparison of the temporal swimming patterns of two real fish, two simulated agents, and real-virtual fish pairs in interactive virtual reality.** The normalized correlation function $C_{ij}$ of the beats (see Fig. 1D) as a function of the mean beat timing difference $\bar{\tau}$. **A** Two real fish swimming together in the same arena. **B** Simulation of two interactive agents using the model with $\beta = 0.3$ for the stochastic parameter produces similar temporal patterns as for 2 real fish. **C** Results for real fish swimming with an interactive virtual fish whose reciprocal behavior is controlled by the model. Due to the small but inherent noise (and delay) in the VR system, here $\beta$ was set to 1 to avoid adding additional noise. The curve in blue shows the actual correlation function, whereas shuffled pairs generated by randomization (randomizing the entire dataset 100 times, resulting in 100 different curves; see the methods section for more details) are in yellow.

We find that unidirectional information flow is insufficient to allow A to achieve an out-of-phase coupling with B, and instead—as in our experiments with unidirectional information flow—A will tend to adopt a fixed, constant, temporal lag with respect to B, irrespective of the frequency of bursting exhibited by B (Fig. 2E). Thus, our experimental results under conditions of unidirectional information flow are consistent with the proposed model.

If we allow bidirectional information flow between A and B in the model, however, we find that this alone can recover the observed out-of-phase coupling of bursts exhibited by real fish pairs (Fig. 3A, B). This suggests that bidirectional interactions are required in order to facilitate the temporal coupling observed in real fish pairs.

In order to test this hypothesis, we also evaluated our in-silico model, *in-virtualis*, employing the dynamical social clamp approach to allow reciprocal (closed-loop) interactions between a real fish and a virtual conspecific, the latter employing our above-proposed model to coordinate its bursting dynamics. We found two significant peaks in this correlation function, akin to what appears in real fish pairs and the model (Fig. 3C; Kolmogorov-Smirnov test comparing to randomization: $p < 0.001$). Thus, our model—despite its simplicity—captures the essential features employed in zebrafish temporal coordination. Reciprocal coupling is necessary, and sufficient, to observe the temporal dynamics exhibited by pairs of individuals.

## Functional consequences of temporal coupling

While much has been revealed regarding spatial coupling among individuals in groups, such as that directional information tends to flow unidirectionally, from front to back, but that speed changes flow bidirectionally[14], we know comparatively little about the role of temporal coordination in mobile animal groups. It is apparent, from our above analysis, that individuals are incapable of responding, in terms of timing, to bursts exhibited by a partner if they occur very shortly (<0.1 s) after their own burst. Whether the rules of thumb exhibited for temporal coordination impact the responsiveness of individuals to changing social conditions, such as turning by their partner, over longer timescales is unknown.

To test the possible functional consequence of temporal relationships between a leader and a follower swimming in close proximity, we investigated how a followers' responsiveness to a sudden change of direction (a turn of 60 degrees) of a (virtual) partner (exhibiting multiple fixed inter-burst intervals, Fig. 2C) is impacted by the temporal relationship between them, controlling for spatial factors

like proximity, or degree of alignment, prior to the turning event (Fig. 4A, Supplementary Figs. 14 and 15, and see Methods for details).

We found that individuals were considerably more responsive to the direction change of their virtual partner, and thus able to maintain close spatial proximity to them, if, prior to the turn, they exhibited the specific temporal relationship that we found in the open loop experiments (Fig. 2D). By comparing turning events with and without the specific temporal relationship, we found significant differences in the distances between the RF and VF for time intervals shortly after the turn (proportion test: $-0.5\ s < t < 0\ s$: $p = 0.1578$, $0\ s < t < 0.5\ s$: $p = 0.0010$, $0.5\ s < t < 1\ s$: $p = 0.0010$, $1\ s < t < 1.5\ s$: $p = 0.0020$, $1.5\ s < t < 2\ s$: $p = 0.0001$; $N = 423$ for the specific, and $N = 144$ for the nonspecific temporal relationship cases) (Fig. 4B) and while before the turn pairs were swimming in close proximity, the success of staying in close proximity was shown to not be influenced by spatial factors prior to the turn (Supplementary Fig. 15).

We finish by returning to our observational data of pairs of RF, and ask whether different temporal coupling regimes might be associated with certain spatial configurations, and whether such spatial configurations may also impact information flow/influence. We find that when their coupling is approximately out-of-phase, they tend to swim side by side (Fig. 4C and Supplementary Fig. 16; projecting the 2D distributing in each panel on a circular axis, and then computing a circular Kuiper test, we obtain $p = 0.001$). We note that it has been suggested that such side-by-side swimming may indeed be beneficial for social influence. Firstly, it facilitates reciprocity of information flow (any other configuration would be asymmetric in this respect), and furthermore, based on geometric principles, it has been shown that this configuration can allow individuals to optimize their detection of both speed and heading changes of a partner by utilizing perceived angular velocity and loom (approaching/receding) on the retina, respectively[60].

## Discussion

Analysis of collectives has proven to be a daunting task, with a myriad of interactions all happening simultaneously. Here we zoomed in to focus on pairwise interactions, with the hope of elucidating their dynamics as a starting point for understanding emergent collective-level outcomes. This in itself is not trivial; reciprocal feedback makes establishing causal relationships of social influence challenging. Non-reciprocity, in the physical sense (the force exerted by one body on another wouldn't be reciprocated equally), has recently been

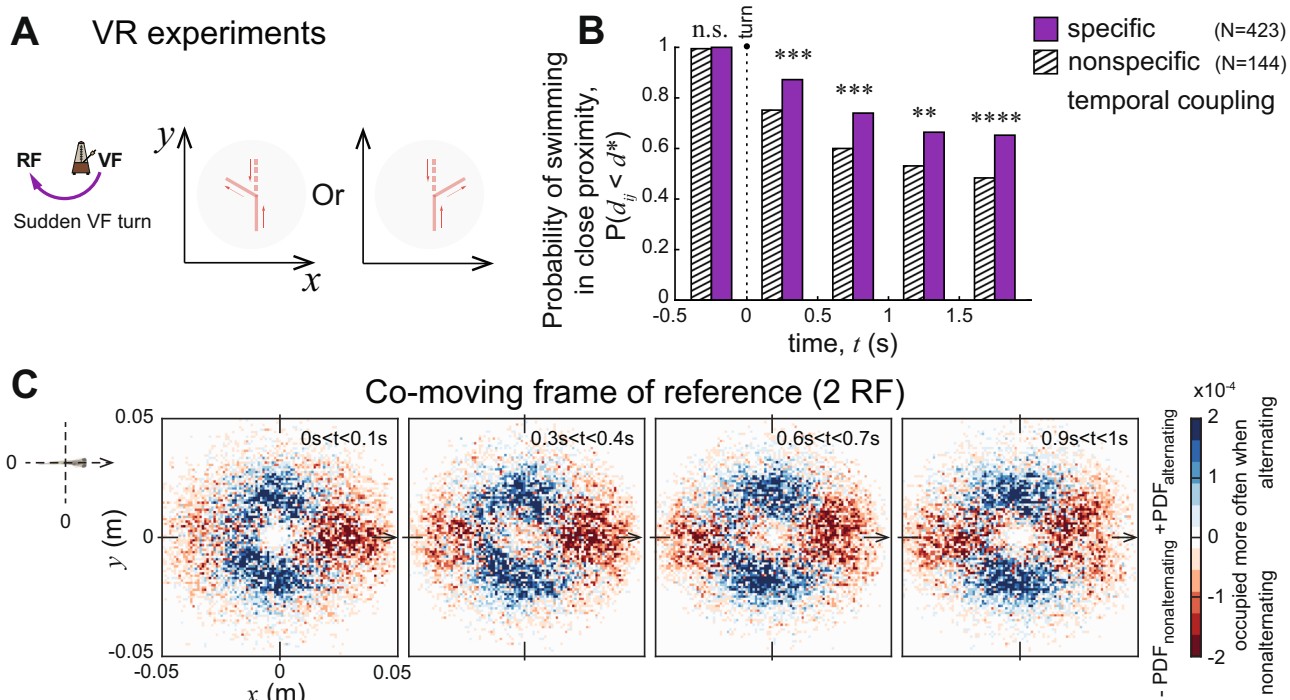

**Fig. 4 | The functional significance of temporal coupling. A** Illustration of the VR experiments when the VF performed a sudden, unexpected turn to a randomly chosen direction (left or right; ±60°). **B** Bar plot shows the probability of being in close proximity ($d < 0.04$ m) at different times right before and after the VF turned at $t = 0$. We analyzed cases where the fish was swimming close to the virtual conspecific prior to the turn. Such turning events were assigned to two categories based on the temporal pattern between the fish before the turn. One category was for cases where the fish followed a specific temporal coupling with the VF (shown with purple), with a peak in the correlation as we previously detected (Fig. 2D); in the other category there was no peak in the correlation close to what was previously detected (nonspecific temporal coupling; hatched bars) (see Supplementary Fig. 14 for the full dynamics of all the data). Those with the specific temporal coupling had a higher likelihood to continue swimming with the virtual fish after it turned. (n.s.

shown for not significant; * for $p < 0.05$; ** for $p < 0.01$; *** for $p \leq 0.001$; **** for $p \leq 0.0001$; exact results from a one-sided proportion test: $p = 0.1578$, $p = 0.0010$, $p = 0.0010$, $p = 0.0020$, $p = 0.0001$). **C** Heatmaps of the spatial configurations between 2 RF, shown in a co-moving frame of reference that follows the smoothed motion path of the focal fish to decrease the jittering effect of its speed oscillations caused by the burst-and-glide pattern (see Methods for more details). Swimming in close proximity was separated into two categories based on the temporal relationship of the two fish: time segments when the pairs had a specific temporal relationship of alternating beats and those that did not. The heatmaps show the difference between the two distributions, where blue depicts positions occupied more often when the pair was alternating, and red shows positions occupied more often when nonalternating.

highlighted as a key ingredient in out-of-equilibrium systems[61]. Although behavioral reciprocity isn't strictly equivalent, here we provide, previously unattainable, evidence of how a real system operates in this regard.

Taking inspiration from the study of coupled-oscillators, we mapped the PRC of our system (a graphical method that was, to the best of our knowledge, never used in the context of pairwise movements). Owing to this, we were able to explicitly detect an unresponsive window of time in the burst-and-glide cycle of the fish, and to derive a simple linear mode of response in the responsive window. Due to these direct measurements, we were able to avoid unnecessary a priori assumptions in our modeling. Our confirmation of the plausibility of the model is twofold—as two agents interacting in-silico, and as a novel hybrid, of essentially the same agent (now rendered as an avatar virtual fish) interacting with real fish.

We found that coordinated motion in zebrafish relies on two-way coupling. This is in contrast with other animal collectives displaying coupled oscillator dynamics such as certain firefly swarms, where one-way interactions are sufficient—a firefly can entrain to a periodic artificial light[62]. One reason for this difference could be the fact that these fireflies are relatively isochronous, which can enable a nonreciprocal agent (not influenced by the firefly) and the firefly to fall in step. Bidirectional interactions have their possible advantages, though. It allows both parties to reach a consensus over the selected frequency at which they operate. Therefore how synchronization (or anti-

synchronization) emerges in nature can take a variety of forms, and finding where common solutions/algorithms are utilized across the tree of life (or not, and the reasons thereof) is of importance.

We performed extensive analyses of our model. By adding an additional parameter $\gamma$ to our model, we demonstrate a continuous transition from one- to a two-way interaction. With further analysis, we also showed that the model is robust to specific parameter values and will work also with a wide range of different slope values, with noise playing a vital role here—the added stochasticity stabalizes the system.

We also provide evidence from VR experiments that temporal coupling influences spatial responsiveness. Further suggestive evidence comes from analysis of real fish pairs, which revealed substantially different spatial configurations at times of different temporal coupling regimes. These findings open up new theoretical and experimental directions of inquiry. While, to date, many modeling studies have focused on metric vs topological interactions, here we suggest that the interplay of spatial and temporal factors play an important role, and much is left to fully understand how this manifests itself in entire collectives.

Another theoretical avenue that has gained attention of late, is so-called Swarmalator systems[63-65], whereby collective motion is combined with temporal coupling. Our results offer an example of such a system in the real world, and crucially, how it may differ from the proposed toy-models. As mentioned, in our case the internal oscillator is the speed of the agent (fish), the oscillation is

non-isochronous, and the coupling tends to be in an alternating fashion.

Lastly, the application of VR environments opens up powerful opportunities for gaining insight into the dynamics of social influence underlying collective behavior. As mentioned, this direction already bore fruit in the study of human interactions[36]. The application of various experimental methods (robotics, VR, etc) that enable researchers to mimic animal conspecifics can pave the way for a much deeper understanding of coordination dynamics. An obvious next step would be to utilize these experimental paradigms with zebrafish in an attempt to understand the neural underpinnings of temporal coupling dynamics.

## Methods
### Fish
All experiments were conducted on 1 cm ± 0.1 cm long zebrafish of age 24 to 26 days post-fertilization raised in a room at 28 °C on a 12 h light, 12 h dark cycle (light switching on and off and 7 am and 7 pm). The fish were bred and raised by the animal care staff of the Department of Collective Behavior at the Max Planck Institute of Animal Behavior and the University of Konstanz. Fish were transferred to the experimental room at least 12 h prior to the experiments in water from their holding tanks. This ensured that the water quality in the experimental room was the same as in their holding facility. This water was also used in the experimental setups (either the arenas for 2 RF experiments or the fishVR setup for virtual reality experiments) where water changing was done once a day. All the fish were naïve, and chosen at random from their holding tanks. All experiments were conducted in accordance with the animal ethics permit approved by Regierungspräsidium Freiburg, G-17/170, G-17/46, and G-21/135.

### Fish length measurement
Fish length was measured by recording them using a custom-built set-up where we mounted a camera above a petri dish. Using the TRex software[39], we measured the length of the midline of the fish.

### Experimental conditions
Room temperature was kept at 26 °C. Experiments were carried out during the day, between 07:00–19:00.

### Pairs of RF
We tracked pairs of juvenile zebrafish in two experimental conditions: (A) a 30 × 30 cm square arena or (B) in petri dishes of 28.7 cm diameter (circular arena). In both conditions, water was filled to ~0.5 cm depth, to allow the fish to swim at ease, but confined to a quasi 2D environment. The fish were filmed from above at 100 fps with Basler cameras (acA2040-90um 2K NIR) and lit from below with an array of infrared lights. The set-up was lit from above with visible light to allow the fish to see each other and the environment, with DÖRR DLP-2000 LED. After being transferred to the arena, each individual first got 5 min for acclimatization (in condition A), and 20 min (in condition B). In condition A, we executed 10 sessions with 10 different pairs, each for 55 min, between 6.12.2017–22.01.2018. In condition B, we recorded 19 pairs, each for 100 min between 7–14.09.2020. The fish were tracked using the TRex software[39]. To reliably detect the fish, we used an intensity threshold of 22 (on a gray scale from 0 to 255; to detect only objects that had similar brightness as the fish) and blob sizes in the range of 0.001 to 1 (to filter for specific object sizes which are around the size of the fish). The videos were cropped for the size of the arena.

### VR experiments
Experiments were conducted in a VR setup produced by loopbio GmbH. One can refer to Stowers et al. (2017)[35] for the details of the fishVR setup. Fish were tested in a spherical cap-shaped acrylic bowl of a maximum of 34 cm diameter. After transferring to the test arena,

each individual first had some acclimatization time, and then the VR stimulus began (a detailed description below). The VF model (its visual appearance) that we used in all experiments is the same as in Stowers et al., 2017[35]. In all VR experiments, only a single VF was presented.

### VR - Nonreciprocal VF
In total, 74 individuals were tested in the no-turn experiment, and 77 in the turn experiment (the recording in a few trials failed). Each fish first had 5 min of acclimatization time. Then the experiment with 1 VF with no turns (no perturbations) occurred for 30 min, followed by 30 min of the experiment with the VF turns. Experiments were conducted between 23-30.5.2020.

The VF appeared in a random position within the bowl, 12 cm away from the planar (horizontal) centroid, oriented to face the centroid (with 0° pitch and roll). The VF was situated 3 cm below the surface of the water as in this region the projection is optimal and this allows for a long duration of swimming (compared with deeper positions due to the curvature of the bowl). The VF swam in a straight line through the 2D centroid of the bowl at that height. The speed of the VF was pre-defined according to fish kinematic statistics (Fig. 1B, D). We then generated typical burst and glide patterns with different frequencies, keeping the integral equal, so the average speed in the different frequencies was kept constant (Fig. 2C). After swimming for ~6 s it turned back to move along the same trajectory, but in the opposite direction (a 180° turn occurred when the VF reached a minimum of $v$, thus causing small variation in the exact duration of a straight path segment for the different frequencies). After a few sec (up to 1 min, depending on the experiment—explained below), the VF would disappear and reappear after a short break (5–15 s; randomly chosen duration) in a new random position, with a new random frequency (out of the five values used in this study).

- with no VF turns: In these experiments, the VF swimming continued for 1 min after which the VF disappeared and reappeared in a new position and frequency.

- with VF turns: These experiments are equivalent to the experiments with no turns (and were carried out subsequently), except for the following differences: after the first straight path segment followed by the 180° turn, the VF turned in a randomly selected direction (±60°; either left or right). The turn occurred in close proximity to the center of the bowl, timed at the VF's tailbeat (thus slightly different for each frequency; the timing after the VF initially appeared was: $T = 240$ ms, $t_{turn}$=9.6 s; $T = 270$ ms, $t_{turn} = 8.91$ s; $T = 300$ ms, $t_{turn} = 9$ s; $T = 330$ ms, $t_{turn} = 8.91$ s; $T$=360 ms, $t_{turn} = 8.64$ s). After the turn, the VF continued to swim in the same direction through the bowl and beyond, as the VF can, in principle, be projected to any position in space, even outside the borders of the bowl. The VF then disappeared and after a short break, a new frequency and location were selected as described for the no VF turn case.

### VR - Reciprocal VF
These experiments were conducted between 05.02.2024–13.03.2024. In total, 67 individuals were tested. Individuals had 20 min for acclimatization, followed by 100 min of the actual VR experiment. The duration of the burst and glide of the VF was dynamically controlled according to the PRC rule or randomly selected from pre-collected periods of real fish (see below). The detailed burst-and-glide patterns of VF were controlled with a piecewise function:

$$v = \begin{cases} a \cdot t + b & \text{if } t \leq 0.12 \\ e^{c \cdot t + d} & \text{otherwise} \end{cases} \qquad (1)$$

Where $v$ is the speed of the virtual fish at time $t$, $a = 0.88$, $b = 0.188$, $c = -10.5$, and $d = -0.823$ are fitted parameters. $t = 0$ represents the start of the period. Note that the start of the acceleration can happen

when the VF has different instantaneous speeds. In general, when the virtual fish accelerates (its burst), it is by 0.88 m/s$^2$.

According to the PRC we evaluate the next bout duration of the VF, namely when to start the next cycle with a burst. The bout duration of virtual fish is determined based on the time difference between the burst time of the VF and the burst time of the RF, $\tau$.

$$T = \begin{cases} \mathrm{rand}(f_0 > 100\,\mathrm{ms}) & \text{if } \tau < 100\,\mathrm{ms} \\ 2\tau & \text{if } 100\,\mathrm{ms} \leq \tau \leq 400\,\mathrm{ms} \\ \mathrm{rand}(f_0 > 400\,\mathrm{ms}) & \text{otherwise} \end{cases} \quad (2)$$

### Data analysis
Data processing and analysis were done with MATLAB versions 2020a and 2023b.

### Speed calculation and filtering
The instantaneous speed was calculated using a five-point stencil method, then smoothed with a Savitzky-Golay filter with a window of 11 frames (=110 ms) and a polynomial order of 2.

### Minima detection
The minima of the speeds (the start of each tailbeat) were detected using the find peaks function in MATLAB on $-v$. Minima that were above 5 cm/s were discarded as outliers. Also, minima that weren't followed by a pronounced acceleration (100 ms after the minima the increase in speed was <2 cm/s) were also discarded to only account for actual bursts and not small variations in speed.

### Temporal coupling correlation
Time windows of 100 frames (1 s) were extracted for each fish. In these times the fish had to be close to each other (Euclidean 2D distance < 4 cm for at least 0.75 s). The cross-correlation was calculated on a discretization of the speed, which is the timing of the minima (the timing of the bursts). This was calculated according to $B \in \{0, 1\}$ where $B = 1$ when $\dot{v} = 0$ and $\ddot{v} < 0$, and $B = 0$ otherwise. Note that each $B = 1$ was cushioned with a Gaussian window, according to

$$w(n) = e^{-2(\alpha n/(L-1))^2} \quad (3)$$

Where $L$ is the window length of 5 frames (=50 ms), and $\alpha$ is the width factor of 2.5. The correlation function then reads

$$C_{ij}(\tau) = B_i(t) \cdot B_j(t + \tau) \quad (4)$$

We present the sum of all these cases. Due to the symmetrical relationships between the two RF, we treated all these fish as the focal fish.

The autocorrelation was calculated similarly to the cross correlation between two fish, only by using the fish with itself, that is

$$C_{ii}(\tau) = B_i(t) \cdot B_i(t + \tau) \quad (5)$$

### Statistics for the correlation function
We shift the correlation function $C$ by subtracting it from its mean, more precisely

$$C_{\mathrm{shift}} = C - \langle C \rangle \quad (6)$$

afterwards, we calculate the integrals of the resultant peaks in $C_{\mathrm{shift}}$.

### Co-moving frame of reference
We calculated the relative position of the neighbor similarly to[14,66], where the focal fish is at the origin of this moving coordinate system (and in this case, the focal fish is facing east). To achieve this we applied translation and rotation transformations on the original coordinate system that was fixed to the environment (i.e., the bowl). By doing this, we can investigate where the neighbor is relative to the focal. To account for the burst-and-glide motion of the focal fish, which would result in bursty tracks even for a neighbor swimming on a straight path with constant speed, we implemented one addition, that the co-moving frame of reference followed the mean motion of the focal fish within each 1-sec-long time window (see Supplementary Fig. 16 for the full visualization). More precisely,

$$x_{new}(t) = x(t) - v_0 t \quad (7)$$

where $v_0$ is the mean speed calculated from the first and last positions of the focal in this 1 s time window.

### Shuffling
Randomizations were generated by shuffling the trajectories of the individuals—pairing individuals (whether real or virtual) that hadn't participated in the same experimental trial (pairing between a focal individual and another individual randomly selected from another pair) and treating them as if they had been swimming together. These randomized trials were analyzed identically as real pairs. For each case, we performed 100 randomizations.

### Statistics
For the proportion test, to calculate the test statistic we used the following equation

$$z = \frac{P_A - P_B}{\sqrt{\frac{P_A(1-P_A)}{n_A} + \frac{P_B(1-P_B)}{n_B}}} \quad (8)$$

Where $P$ is the proportion of values for each specific case. We then determine if the result is significant (the p-value) with the cumulative distribution function (CDF) of the standard normal distribution, evaluated at $z$.

All statistical tests, where appropriate, were two-sided.

### Handling errors in tracking and VR
To account for errors in detection from the tracking algorithm of the pairs, we omitted from our analysis all data with undefined x, y, or speed values. For the VR experiments, we omitted cases where we detected errors in the VR output, whereby the speed of the VF wasn't according to what we had assigned. That is, if we detected too low or too high speeds (<0.001 m/s or >0.11 m/s) for more than 5 frames within our window of analysis (100 frames in the case of the nonreciprocal VF with no VF turns and 300 frames in the case with the VF turns). In the case of the reciprocal VF, where the speed profile isn't fully determined prior to the experiment, we omit cases according to extreme values of VF acceleration (numerical differentiation of the speed)—that is, if we had more than 5 frames of >0.05 m/s$^2$ or <−0.05 m/s$^2$ in a 100 frame window. Overall the ratio of values omitted, over the total number of values (omitted cases and used cases) was 0.25 for the nonreciprocal VF and 0.14 for the reciprocal VF experiment.

### Simulation
The simulations were designed to test the effect of the parameter settings and the directionality of the information flow on the temporal coupling in our model. We only concentrated on the temporal aspect of the behavior, so movement in space was not implemented.

We ran one- and two-way interactions. In the two-way scenario, both oscillators behaved responsively according to the model described in Rules 1–3, with $\beta$ capturing the level of responsiveness. In case of $\beta = 1$ they perfectly follow the PRC response, and in case of $\beta = 0$ they choose their timing entirely randomly (without taking into account the timing of their neighbor).

In the one-way scenario, one of the oscillators behaved responsively as described above, with the other oscillator ignoring the timing of its neighbor (either acting as a metronome or with random oscillation). We added Gaussian noise with a standard deviation $\sigma = 3$ (in simulation steps) to the responsive oscillator in both the one- and two-way cases. In all models, one timestep is analogous to 0.01 s in the experiments.

Supplementary Figs. 6–9 and 12, 13 show results for two oscillators in multiple scenarios. For each case, we ran 20 realizations, each for 101,000 timesteps, where we analyzed the last 100,000 timesteps, to account for the transient dynamics. Supplementary Fig. 10 is a parameter space exploration of the transients to assess the influence of the initial conditions (the initial lags) on the dynamics. Supplementary Fig. 11 is an exploration of varying degrees of interactivity. In both cases, we looked at the phase differences after 1000 timesteps. We ran 1000 realizations of each model for every combination of the parameter values.

### Reporting summary

Further information on research design is available in the Nature Portfolio Reporting Summary linked to this article.

## Data availability

The data that support the findings of this study are available in figshare with the identifier 10.6084/m9.figshare.c.7123501.v1. In addition, source data are provided in this paper in https://doi.org/10.6084/m9.figshare.25626990.v1.

## Code availability

Codes for experiments, simulations, as well as analysis were written either in MATLAB versions 2020a and 2023b (MathWorks Inc., Natick, MA, USA) or Python (Python Software Foundation, 2018). All codes that support the findings of this study are available on figshare with the identifier https://doi.org/10.6084/m9.figshare.25398523.v1

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

## Acknowledgements

We are grateful to the animal care staff at the University of Konstanz, including Christine Bauer, Jayme Weglarski, Dominique Leo, and Alexander Bruttel for help in conducting the experiments, as well as Laura Schröder for her assistance. G.A. acknowledges the International Max Planck Research School for Organismal Biology for the graduate school community and access to courses and resources. M.N. acknowledges the MTA-ELTE Statistical and Biological Research Group, Eötvös Loránd University, and I.D.C acknowledges the Rothschild Distinguished Fellowship from the Isaac Newton Institute for Mathematical Sciences as well as the Max Planck Society. The following authors acknowledge the following grants: - Hungarian Academy of Sciences Grant 95152; M.N. - The Hungarian National Research, Development and Innovation Office grant no. K 128780; M.N. - Isaac Newton Institute for Mathematical Sciences for support and hospitality during the programme 'Mathematics of Movement: an interdisciplinary approach to mutual challenges in animal ecology and cell biology', supported by the EPSRC Grant Number EP/R014604/1; M.N. and I.D.C. - Sino-German mobility grant M-0541; L.L. - Messmer Foundation Research Award; L.L. - Office of Naval Research Grant N00014-19-1-2556; I.D.C. - Horizon Europe Marie Sklodowska-Curie Actions (860949); I.D.C - Struktur- und Innovationsfonds für die Forschung of the State of Baden-Württemberg, the Deutsche Forschungsgemeinschaft (German Research Foundation) under Germany's Excellence Strategy EXC 2117-422037984; I.D.C. - The PathFinder European Innovation Council Work Programme #101098722; I.D.C.

## Author contributions

G.A., M.N., and I.D.C. conceived the idea and designed the project; G.A., L.L., M.N., and I.D.C designed the experiments; G.A. and L.L. implemented the experiments and collected the data; G.A. and M.N. analyzed the data; G.A. and M.N. devised, implemented and analyzed the model; G.A., M.N., and I.D.C. wrote the paper with feedback from L.L.

## Funding

## Competing interests

The authors declare no competing interests.
