## [Peer Review File · Nature Communications]

REVIEWER COMMENTS

Reviewer #1 (Remarks to the Author):

In this manuscript, the authors performed simulations and innovative experiments to investigate the temporal aspect of the coupling between zebrafish pairs during their collective motion. They found that bidirectional coupling is essential to explain the "out-of-phase" dynamics observed experimentally and that such temporal coupling can help fishes stay close together after abrupt direction changes. The paper is well-written and interesting, and the VR experimental setup can be useful for many other studies. Before I can recommend publication, however, there are two issues that I would like the authors to address:

1. One crucial piece that is missing in the current manuscript is a theory explaining why bidirectional coupling can produce "out-of-phase" dynamics while one-way coupling cannot. It seems that the model is simple enough that some kind of dynamical systems analysis can be done. For example, showing that the "out-of-phase" state is unstable for systems with one-way coupling.
2. Is there a reason why the relevant data are only available upon request and not made available online upfront? This creates unnecessary friction for other groups to reproduce and extend this work.

Reviewer #2 (Remarks to the Author):

This paper addresses fundamental questions about the synchronization of fish/collective movement. It's a real pleasure to read this excellent and original article. Indeed, addressing the problem of collective movement by focusing on the coordination of two units seems to me to be an original idea and a useful approach.

These results are of sufficient general interest to justify a publication in Nature Communication because through the case study, the experimental and theoretical methodology and the hybrid system (interaction with "natural fish"). I am convinced that the present work can be a source of inspiration in other fields, including physics or robotics and can be extended to many other questions and situations. Indeed, these problems (synchronization, rhythms) are the focus of attention in many (biological) fields and moreover, I share the authors' opinion that these issues of 'coupling' in the organization of collective movements have been neglected (Lines 62-64).

Unsurprisingly, I have no major comments, just a few minor suggestions, mainly to attract attention from other areas.

Introduction

The introduction sets out the framework and objectives of the document and is not aimed solely at social fish specialists.

Lines 113-115 “in most natural fish populations [28], and that it has been found that even when schooling individuals tend to swim close to, and behaviourally couple most strongly with, a single neighbour [37]”: Could your results (or model) provide an explanation to this phenomenon?

Line 160 “As is common among freshwater fish “. I'm not familiar with the biology of fish: this is not the case for marine (saltwater) species? Why?

Results

Line 175 Related works other than in neurobiology? E.g. Shahal et al 2020, Nature Communications ?

Line 207- 215 “We note that regular temporal dynamics at a collective level need not imply strong, or any, rhythmic behavior on the part of system components....Sustained rhythmic coupling between system components can, for example, emerge spontaneously when these components mutually excite one another (such as a neuron's firing increasing the probability of another to whom it is connected to subsequently fire), but also exhibit a refractory period...”

I totally agree and it is not sufficiently taken into account in the biology of societies, although it has been studied in other areas of biology (biochemistry, ecology). There are some works in 'social biology' (e.g. ants : Cole The American Naturalists, 1991, Franks et al Bulletin of Mathematical Biology, 1990; spiders Krafft & Pasquet, Insectes Sociaux, 1991, Chiara et al, PNAS, 2022).

Your work is of great interest to many areas of social biology, not just studies of shoals and collective movements. Your results may stimulate work in these areas.

The positive feedback-refractory period pair is undoubtedly one of the basic templates for generating collective rhythms, not only in neurons but also in many domains including social systems (e.g. the toy model of Deneubourg & Goss, Insectes Sociaux, 1988; Chiara et al, PNAS, 2022);...

Discussion

Lines 577-581 Are you suggesting that there is a critical value in the determinism of the rhythm for transiting between "one-way" and a "two-way coupling"? I don't know if this has ever been demonstrated. If not, it would be interesting to explore this in the future.

In conclusion, I strongly recommend this outstanding paper for publication

Reviewer #3 (Remarks to the Author):

The paper “Revealing the mechanism and function underlying pairwise temporal coupling in collective motion” by Amichay et al. looks at pairs of juvenile zebrafish swimming in the same arena

and tries to understand their swimming patterns in terms of the temporal correlation of swim events between animals.

Overall, I find the study potentially interesting as it addresses a general problem. I do find however that the text and the statistics are sloppy in places, and that there are significant references missing to work by Dreosti et al and Larsch et al. Improving the statistical (and scientific) rigor is important before considering this manuscript for publication in a journal such as Nature Communications.

Comments:

(important ones are labeled with an asterisk)

Line 59: has instead of have

Line 70: Evidence has been found that: add references

Line 85: Add reference to Odstroil et al., Current Biology

Line 93 – 99: What does this mean? Movement-triggered sensory cross-correlation has been used to deduce sensory filters across many sensory modalities.

Line 102: Add reference to Larsch and Baier, 2018.

Line 136-138*: The authors should show error bars or ideally confidence intervals in B, D etc.

Line 168-170*: Can the authors show the PDFs for the far and randomized cases. Maybe change the word lag to interbout times or bout frequency.

Line 173-175: can the authors add references to this.

Line 200*: What is beta in this experiment? How was it chosen?

Line 203: Why is the maximum speed also changed? The figure suggests that this is to keep the average velocity constant, but why is this important?

Line 204: I do not understand the sentence: Purple bands show the range that these peaks occurred in which defined a “shared lag”, I feel it is grammatically unclear.

Lines 171 – 215: These paragraphs seem strange in the middle of a results section and in the middle of discussing a figure: it is unclear what their point is.

Lines 235 – 237*: Given that 1E does not show any notion of variance, the definition of an unresponsive temporal window (whether animals are indeed unresponsive) is unclear to me: why does 1E left look different from 1E right then?

Lines 243 – 247***: Can the authors perform linear regression and report the slope of the regression line and the confidence intervals? This should be done for each fish and then the results averaged across fish. The statistics shown in S5 test something totally different: how well $y=x^2$ fits the data compared to how well it fits random data, which is very different.

Lines 245 – 246: The biological importance of the relation $y=x^2$ should be more clearly explained because this is the first time that it is mentioned and it is very important.

Line 256: What is the distance between the VF and the RF? The authors showed in Figure 1 that the behavior depends on this distance, but this is now disregarded.

Line 256: The authors claim that “This finding shows that unidirectional information flow is insufficient to facilitate temporal coupling.” What is beta here? I think it would be convenient for the authors to describe explicitly what they mean by temporal coupling: Is it reproducing Figure 1E left?

Line 291*: How is this beta determined? Can the authors quantify the behavior of their model as a function of this free parameter? Supplementary figures are included, but these are not mentioned at all and it is very unclear what is important about $\beta = 0.3$ or whether this is the case for any “social coupling”.

Line 319: Can the authors plot this for $\beta = 0.3$? If the system is too noisy (or there is too much delay), then the system cannot really be used for experiments.

Line 326: Substitute “we also evaluated our in-silico model, “in-virtualis”,” by: we ran simulations.

Line 331*: Can the authors display confidence intervals?

Line 349 - 350: The statement “if, prior to the turn, they exhibited a specific temporal relationship akin to what we found in the open loop experiments” is very vague: can the authors be more explicit please.

Lines 381 – 383: I do not understand what these two categories are.

Line 384: Can the authors include confidence intervals and explain how they performed the proportion test.

Line 396: “which is a more stable configuration—both for reciprocity, and also in terms of visual sensitivity and detection” is an interesting observation; can the authors expand on this please? Referring to [52] is good, but if the authors mention it, it should be clear to the general reader why this is the case.

Line 540: These methods “Co-moving frame of reference” should be expanded upon as they are currently insufficient to understand what was done in detail.

Line 545: Could the authors provide their simulation code?

Line 566 - 567: What is the point of this statement regarding Newton’s Law? Do the authors want to define what non-reciprocity is or what is an unbalanced system (in the sense of having a resultant force acting upon it)?

Line 577 – 579: What about other fish swimming or vocalizations? Comparing juvenile zebrafish to fireflies (only) seems slightly random.

Reviewer #4 (Remarks to the Author):

Thank you for the opportunity to review the manuscript titled “Revealing the mechanism and function underlying pairwise temporal coupling in collective motion”. In this work, the authors address an intriguing, and to my knowledge, previously unexplored research question: How do animals in collective motion time their actions with respect to one another, and what functional implications does this temporal coupling bear? By employing an impressive combination of experiments, models, and data analytics, the authors present findings that will be of interest to a broad audience. Primarily, they discovered that schooling zebrafish larvae tend to alternate the timing of their tailbeats while swimming in pairs. They also introduce a straightforward rule explaining this synchronization, emphasizing that temporal coupling between two schooling fish emerges only when both adhere to this rule. The authors also demonstrate the functional significance of these behaviors, demonstrating that temporal coupling allows fish to avoid sensory limitations that would manifest if two fish were to accelerate simultaneously.

This research is compelling and promises to spur further work on the implications of temporal coupling on collective behavior in diverse contexts and scales. Nonetheless, I have several reservations that require attention:

1. Concerning beta:

- The statement: “We find that, as in our virtual reality experiments with unidirectional information flow (Fig. 2D), that for relatively weak social coupling, and correspondingly relatively high intrinsic stochasticity (β), that unidirectional information flow is insufficient to allow A to achieve an out-of-phase coupling with B” is confusing because the authors set $\beta = 0.3$, which would seem to imply the opposite (they follow social cues more often than not).
- In the Supplementary Information, for example, Figure 7, the effect of changing beta is the opposite of what I would expect. $\beta = 1$ should correspond to completely random tailbeat timings, yet the bottom right panel shows the opposite. Perhaps the given definition of beta was flipped?
- The manuscript would benefit from justifying the selection of beta in the main text. If the necessity of two-way information flow for temporal coupling is contingent on the beta value, then it's crucial to

use a beta value consistent with Panel 1E. Visually, the data seems better represented by strong social coupling. A potential approach could be estimating beta by minimizing the KL divergence of the modeled and observed bout duration distribution as a function of tau.

- The main text should also incorporate a discussion on the implications of varying beta. How does the strength of social coupling affect temporal coupling in scenarios of both one-way and two-way information flow?

2. Presentation clarity:

- The interpretation of certain results could be made clearer. For instance, lines 173-174 state, ""fish are found to exhibit prominent temporal coupling with respect to the timing of their bursts, exhibiting a characteristic (shared) time-lag, and out-of-phase (i.e., alternating) relationship, between bursts." It would be clearer to reference Figure 1C here and further spell out how this is demonstrated by Figure 1C.

- Furthermore, the section "One-way information flow..." would benefit from explicitly stating your expectations for Figure 2D if one-way information flow was sufficient for temporal coupling. For example, something like: "If one-way information flow is sufficient for temporal coupling, we would expect the timing of the focal fish's tailbeats to alternate with those of the virtual fish. Thus, we would expect to see a drop in the correlation function at values of tau near zero, and peaks in the correlation function when tau is equal to half the beat frequency of the virtual fish. Instead, we find..."

- What is the benefit to showing the correlation functions over negative beat timing differences?

3. Intuitive explanation of mechanisms:

- While the authors do an impressive job combining data analytics, models, and VR experiments to demonstrate that two-way information flow is needed for temporal coupling, I am still left wanting an intuitive explanation for why one-way information flow is insufficient for coupling. Given the authors have a simple model, it should be possible to delve deeper into the mechanisms underlying the results.

4. Figure 3:

- There's a noticeable difference between the y-axis scale in panel B compared to A and C. Is this merely due to varying bin sizes, or is there another factor?
- For panel C, it isn't entirely convincing that it captures the essential features in panel A. It would be useful to have a null distribution to compare the line in C to, for example, by a randomization method.
- It might also be worthwhile to discuss why the strength of the pattern appears much weaker in the VR experiment compared to the RF experiments.

Minor comments:

- Line 169: The assertion about the temporal structure in relation to Fig 1D needs validation.
- Figure 1E: A colorbar inclusion would greatly enhance the interpretation.

In summary, this study holds significant potential. With a few modifications, I believe it could serve as a cornerstone in understanding collective motion.

REVIEWER COMMENTS

Reviewer #1 (Remarks to the Author):

In this manuscript, the authors performed simulations and innovative experiments to investigate the temporal aspect of the coupling between zebrafish pairs during their collective motion. They found that bidirectional coupling is essential to explain the "out-of-phase" dynamics observed experimentally and that such temporal coupling can help fishes stay close together after abrupt direction changes. The paper is well-written and interesting, and the VR experimental setup can be useful for many other studies. Before I can recommend publication, however, there are two issues that I would like the authors to address:

We thank the reviewer for the positive feedback and the appreciation of our work, as well as thoughts and comments that helped us improve the work.

1. One crucial piece that is missing in the current manuscript is a theory explaining why bidirectional coupling can produce "out-of-phase" dynamics while one-way coupling cannot. It seems that the model is simple enough that some kind of dynamical systems analysis can be done. For example, showing that the "out-of-phase" state is unstable for systems with one-way coupling.

We agree, this is a good point. We have now added more simulations and analyses of the model to clarify this further. We now have simulations where we analyzed the short-term behavior of the system (transients), when started from specific initial conditions (different initial lags) to see whether the outcomes we observe are stable irrespective of the initial conditions. We compared 4 different versions of the model:

- Both oscillators are interactive, and we vary the level of stochasticity with the parameter β for both (always equally).
- Both oscillators are interactive, but one is fixed at $\beta=0.3$ and for the other we vary β with increments through all its possible range—from 0 to 1.
- Both oscillators are interactive, but one is fixed at $\beta=0$ (not allowing two-way interaction) and for the other we vary β with increments through all its possible range.
- One oscillator is interactive, while the other completely ignores the social input (and follows its internal fixed-rhythm beating like a metronome). This corresponds to the one-way interaction reported in the original version of our manuscript. But to go even further in the understanding we extended the scope to not just study one- or two-way interactions, but a whole range as a continuous variable, where we introduce a new parameter (γ) which controls how socially-responsive the other oscillator is. For $\gamma=0$, the agent follows its fixed beating pattern, and when $\gamma=1$, it is fully interactive and controlled by the phase response curve (PRC), while values between 0 and 1 defines the probability of choosing a beat based on the PRC or using the internal fixed period length.

These analyses show that there is no sensitivity to the initial conditions for the interactive cases (high β values). In case of low (zero or close to zero) values for the interactivity β , then even if we start from an initial condition that matches the out-of-phase coupling (as suggested by the reviewer), the system can quickly advance to any possible time delay and relative beat timing. Varying γ is a new addition where we can study the transition from the one-way coupling (when the partner is metronome-like, similar to the preset VR experiment) to the two-way coupling that enables out-of-phase dynamics. See fig. S10 and S11 at the end of the document or in the appendix of the paper for these results.

In addition, we also analyzed the stability of the model, with regards to the slope of the PRC (m) by looking at its performance with varying slopes. We show that the two-way model is highly robust and will result in out-of-phase relationships for a wide range of values; in the one-way scenario this is not the case. See fig. S12-13.

Attached is an excerpt from the manuscript, explaining the results and discussing them (lines 303-309):

“Note that we studied the behavior of the model with increments of β values over its full range (see SI Appendix, Figs. S6-9 for multiple parameter space analyses). Since our results are robust to this choice (see SI Appendix, Figs. S6-9), we present a single value ($\beta = 0.3$) here (e.g., Fig. 2E), for simplicity, and without loss of generality. In addition, our findings are robust to both the starting conditions (initial lag) (SI Appendix, Figs. S10-11) and, especially in the presence of stochasticity, to the specific value chosen for the slope of the PRC (SI Appendix, Figs. S12-13). This demonstrates that out-of-phase coupling is robust in the face of inherent errors associated with perception and action.”

2. Is there a reason why the relevant data are only available upon request and not made available online upfront? This creates unnecessary friction for other groups to reproduce and extend this work.

We initially thought to make all the data and code open access once the paper is accepted for publication. But now we decided to follow the editorial request and we have uploaded the code. The corresponded texts in the main manuscript now read as (lines 696-702):

“Data availability. The data that support the findings of this study are available in figshare with the identifier 10.6084/m9.figshare.24711132

Code availability. Codes for experiments, simulations, as well as analysis were written either in MATLAB versions 2020a and 2023b (MathWorks Inc., Natick, MA, USA) or Python (Python Software Foundation, 2018). All codes that support the findings of this study are available on figshare with the identifier 10.6084/m9.figshare.24598773”

Reviewer #2 (Remarks to the Author):

This paper addresses fundamental questions about the synchronization of fish/collective movement. It's a real pleasure to read this excellent and original article. Indeed, addressing the problem of collective movement by focusing on the coordination of two units seems to me to be an original idea and an useful approach.

These results are of sufficient general interest to justify a publication in Nature Communication because through the case study, the experimental and theoretical methodology and the hybrid system (interaction with "natural fish"). I am convinced that the present work can be a source of inspiration in other fields, including physics or robotics and can be extended to many other questions and situations. Indeed, these problems (synchronization, rhythms) are the focus of attention in many (biological) fields and moreover, I share the authors' opinion that these issues of 'coupling' in the organization of collective movements have been neglected (Lines 62-64).

Unsurprisingly, I have no major comments, just a few minor suggestions, mainly to attract attention from other areas.

We thank the reviewer for their appreciation of the work, and for the suggestions for making the presentation more thorough and appealing.

Introduction

1. The introduction sets out the framework and objectives of the document and is not aimed solely at social fish specialists.

Lines 113-115 "in most natural fish populations [28], and that it has been found that even when schooling individuals tend to swim close to, and behaviourally couple most strongly with, a single neighbour [37]": Could your results (or model) provide an explanation to this phenomenon?

We thank the reviewer for this interesting comment. The question of why fish couple each time only to a single neighbor and not to multiple is intriguing and important that could shed light on the mechanism underlying collective motion. We wanted to keep the balance in our current manuscript between experimental (both with real fish and with virtual conspecifics) and modeling/simulations. It is possible that the mechanism elucidated in our work may help explain this observed feature of fish schools, but we feel this conclusion cannot be drawn with the available data and so are reluctant to suggest this as a possible reason. However, our work certainly suggests that future studies should incorporate the consideration of temporal *and* spatial coupling in larger groups, as we emphasize in the discussion. This will be a fascinating question for future studies.

2. Line 160 "As is common among freshwater fish ". I'm not familiar with the biology of fish: this is not the case for marine (saltwater) species? Why?

We thank the reviewer to bringing this to our attention; we didn't actually intend to make such a distinction and have therefore rephrased this sentence (lines 158-160):

"As is common among many fish species, individuals exhibit periods of time of highly coordinated motion when relatively close, and uncoordinated motion when relatively far (termed fission-fusion dynamics [28, 38])"

Results

3. Line 175 Related works other than in neurobiology? E.g. Shahal et al 2020, Nature Communications ?

We thank the referee for referring us to this relevant citation, which we added.

4. Line 207- 215 "We note that regular temporal dynamics at a collective level need not imply strong, or any, rhythmic behavior on the part of system components....Sustained

rhythmic coupling between system components can, for example, emerge spontaneously when these components mutually excite one another (such as a neuron's firing increasing the probability of another to whom it is connected to subsequently fire), but also exhibit a refractory period..."

I totally agree and it is not sufficiently taken into account in the biology of societies, although it has been studied in other areas of biology (biochemistry, ecology). There are some works in 'social biology' (e.g. ants : Cole The American Naturalists, 1991, Franks et al Bulletin of Mathematical Biology, 1990; spiders Krafft & Pasquet, Insectes Sociaux, 1991, Chiara et al, PNAS, 2022).

Your work is of great interest to many areas of social biology, not just studies of shoals and collective movements. Your results may stimulate work in these areas.

The positive feedback-refractory period pair is undoubtedly one of the basic templates for generating collective rhythms, not only in neurons but also in many domains including social systems (e.g. the toy model of Deneubourg & Goss, Insectes Sociaux, 1988; Chiara et al, PNAS, 2022);...

We thank the referee to pointing our attention to so many valuable and relevant studies that we missed. We agree that this relates to other systems, and have now added all these citations which we believe significantly improved our introduction of the topic (lines 217-220):

"Similar dynamics are ubiquitous and are found also in animal collectives, such as ants (individuals with temporally "chaotic" activity, that collectively synchronize their activity through mutual excitation) or spiders (which exhibit synchronized movements in pursuit of prey) [46–51]."

Discussion

5. Lines 577-581 Are you suggesting that there is a critical value in the determinism of the rhythm for transiting between "one-way" and a "two-way coupling"? I don't know if this has ever been demonstrated. If not, it would be interesting to explore this in the future.

We thank the referee for this insightful suggestion. We have now performed further modeling to investigate exactly this point. Please see our response above to reviewer #1 comment 1, and fig. S10-11.

In conclusion, I strongly recommend this outstanding paper for publication

We thank you for this recommendation.

Reviewer #3 (Remarks to the Author):

The paper “Revealing the mechanism and function underlying pairwise temporal coupling in collective motion” by Amichay et al. looks at pairs of juvenile zebrafish swimming in the same arena and tries to understand their swimming patterns in terms of the temporal correlation of swim events between animals.

Overall, I find the study potentially interesting as it addresses a general problem. I do find however that the text and the statistics are sloppy in places, and that there are significant references missing to work by Dreosti et al and Larsch et al. Improving the statistical (and scientific) rigor is important before considering this manuscript for publication in a journal such as Nature Communications.

We thank the reviewer for their positive view of our study, and their meticulous reading of the paper, that has helped us much improve it. The papers mentioned are indeed important and relevant so we added them in the current version. Below are detailed replies per each comment.

Comments:
(important ones are labeled with an asterisk)

1. Line 59: has instead of have

We acknowledge the referees careful reading and have corrected this.

2. Line 70: Evidence has been found that: add references

We agree with the referee that these citations were needed here and we added them.

3. Line 85: Add reference to Odstrcil et al., Current Biology

We agree that this is a very important and relevant citation and therefore we have added it.

4. Line 93 – 99: What does this mean? Movement-triggered sensory cross-correlation has been used to deduce sensory filters across many sensory modalities.

We agree with the referee that this aspect has been studied in this way. We have now removed a sentence from this section that we think may ease the reading of this part. It now reads (lines 88-95):

“Thus, despite multiple possible mechanisms having been suggested, the nature and functional consequences of time-varying sensing and motor response in regulating collective behavior remains largely unknown. One of the key issues has been that establishing the time-varying reciprocal coupling of interaction strength among individuals, even when only considering a pair, proves very challenging. For example, motor decisions being made in relatively discrete windows of time, does not inform us about possible windows of perception, or the timescale that informs each discrete motor decision, is obtained [17]. The changing strength and direction of reciprocal social coupling can make the causal time-varying structure of interactions hard to infer.”

5. Line 102: Add reference to Larsch and Baier, 2018.

As per above, this citation is also crucial and we have incorporated it into the manuscript.

6. Line 136-138*: The authors should show error bars or ideally confidence intervals in B, D etc.

We agree with the referee that error bars or some other way of showing the variation in the data would be beneficial. As far as we know, this isn't typically done for PDFs and

correlation functions, but to allow a more thorough comparison we now added more information (histograms, randomizations). Please see figure 1.

7. Line 168-170*: Can the authors show the PDFs for the far and randomized cases.

We added a PDF for the bout duration of the far case as well, as requested. Randomizations were only used between pairs.

Maybe change the word lag to interbout times or bout frequency.

While we agree that interbout times could be used, bout frequency would imply fixed periodicity which isn't the case here. We decided to keep the word "lag" which is commonly used in signal processing contexts.

8. Line 173-175: can the authors add references to this.

We added the following clarification in the text referring to the plot (line 173):

"(as evident in the symmetric curve in Fig. 1C)"

9. Line 200*: What is beta in this experiment? How was it chosen?

β is the strength of the PRC term, spanning from 0 to 1; $\beta=0$ meaning no PRC interaction, so the interbout times chosen entirely randomly to $\beta=1$ meaning no randomness, fully interacting beat timings. We explored β values across its range, and present in the main text figure results for $\beta=0.3$. We do this to show that even when relatively high levels of stochasticity are present (so fish choose to adjust their tailbeats according to the neighbor only 30% of the time, otherwise it ignores it) the two-way coupling is still reproduced (i.e., it is robust).

For the experiment where real fish were interacting with the model through the virtual fish, we chose $\beta=1$ for the model to control the virtual fish, as the VR system already introduces some inherent error (such as small variations of 3D tracking (Stowers et al., 2017) which is accounted for as the noise that is present in the model for lower values of β). A better and more thorough explanation of β is provided now in the paper in multiple sections. Please see below these excerpts (lines 199-200):

"...a random stochastic process characterized by β (the strength of the PRC term, spanning from 0 to 1: entirely random to no randomness, i.e., the latter situation being fully interactive)."

And (lines 303-306):

"Note that we studied the behavior of the model with increments of β values over its full range (see SI Appendix, Figs. S6-9 for multiple parameter space analyses). Since our results are robust to this choice (see SI Appendix, Figs. S6-9), we present a single value ($\beta = 0.3$) here (e.g., Fig. 2E), for simplicity, and without loss of generality"

In the methods section (lines 663-665):

"In case of $\beta = 1$ they perfectly follow the PRC response, and in case of $\beta = 0$ they choose their timing entirely randomly (without taking into account the timing of their neighbor)."

Also refer to our reply to referee #1 comment 1 for other detailed explanations about the function of the model in general, and related comments 18 and 19.

10. Line 203: Why is the maximum speed also changed? The figure suggests that this is to keep the average velocity constant, but why is this important?

We changed the maximum speed so that the mean speed would stay fixed, which allows us to vary only one parameter, the frequency of bouts. In other words, the fish can swim with the virtual conspecific irrespective of the beat timing (that could otherwise cause changes in speed that could cause a separation).

11. Line 204: I do not understand the sentence: Purple bands show the range that these peaks occurred in which defined a “shared lag”, I feel it is grammatically unclear.

We apologize for this misunderstanding and have now changed the text to help clarify this. The caption now reads (lines 204-205):

“Purple bands show the range in which these peaks occurred, which defines the “shared lag”.”

12. Lines 171 – 215: These paragraphs seem strange in the middle of a results section and in the middle of discussing a figure: it is unclear what their point is.

We agree, but this was actually before we referenced our own results there (see our reply to comment 8). Now that that has been added, we think that this section is better off in the results section.

13. Lines 235 – 237*: Given that 1E does not show any notion of variance, the definition of an unresponsive temporal window (whether animals are indeed unresponsive) is unclear to me: why does 1E left look different from 1E right then?

We are sorry if this wasn't clear—a notion of the variance is shown in the right side plot of fig. 1E (the shuffling). We also added a colorbar to enable the reader to fully understand what is shown in each heatmap and to compare the left and right side plots. Please see the updated figure at the end or in the updated manuscript.

14. Lines 243 – 247**: Can the authors perform linear regression and report the slope of the regression line and the confidence intervals? This should be done for each fish and then the results averaged across fish. The statistics shown in S5 test something totally different: how well $y=x^2$ fits the data compared to how well it fits random data, which is very different.

Yes, we thank the referee for the suggestion and agree that such a statistic could accommodate what we've shown. Earlier we wanted to show as pointed out by the reviewer, how $y=2x$ fits the real data as compared to a randomized case. Now we performed a linear regression analysis in addition to what was reported previously (lines 249-250):

“...for each pair separately, we report a mean regression coefficient of 1.71, with $R^2 = 0.48$ ”

While this isn't a very high R^2 value, we have stressed from the beginning that this relationship was approximated for simplicity, and did not intend to derive the exact functional form. Our modeling shows that even with the presence of relatively high noise, the two-way temporal out-of-phase coupling is achieved. Our new exploration of the effect of the slope on the temporal patterns confirms that the exact value of the slope does not matter here (especially in the presence of noise).

15. Lines 245 – 246: The biological importance of the relation $y=x^2$ should be more clearly explained because this is the first time that it is mentioned and it is very important.

We agree with the referee that such a clarification can help the reader understand this point. We now added the following excerpt (lines 252-254):

“In other words, this simple functional form can enable an out-of-phase relationship to occur, even amongst irregular (non-isochronous) oscillators. When one reacts

"early" the other will tend follow suit, or when one is "late" the other will also be inclined to delay its own response."

16. Line 256: What is the distance between the VF and the RF? The authors showed in Figure 1 that the behavior depends on this distance, but this is now disregarded.

We thank the referee for noticing this. The distance between them was restricted to 4 cm (the two fish had to be <4 cm away from each other, for at least 75% of the time in the window analyzed), as in the rest of the analyses in the paper.

17. Line 256: The authors claim that "This finding shows that unidirectional information flow is insufficient to facilitate temporal coupling." What is beta here? I think it would be convenient for the authors to describe explicitly what they mean by temporal coupling: Is it reproducing Figure 1E left?

We thank the referee for this question. β plays no role here when coupling with a metronome (please see also). In addition, we provide another explanation of what we mean by temporal coupling—a peak in the correlation function indicating a tendency for a specific temporal lag in terms of relative beat times. This, as we've learned, can be different in one- and two-way scenarios. See for instance this updated text from the manuscript (lines 265-267):

"in other words, if one-way information flow is sufficient for temporal coupling, we would expect the timing of the focal fish's tailbeats to alternate with those of the virtual fish. Thus, we would expect to see two peaks in the correlation function forming a symmetry about $\tau = 0$ "

18. Line 291*: How is this beta determined? Can the authors quantify the behavior of their model as a function of this free parameter? Supplementary figures are included, but these are not mentioned at all and it is very unclear what is important about beta = 0.3 or whether this is the case for any "social coupling".

We thank the referee for these important questions. β was determined qualitatively but the specific choice of β won't affect the results. Specifically, $\beta=0.3$ isn't a singularity--as we demonstrate with our parameter space explorations. Also, we conducted a thorough analysis of the model and specifically for β (See our reply to referee #1 comment 1). We thank the referee for pointing our attention to the fact that the SI figures weren't adequately referred to and we have now addressed this.

19. Line 319: Can the authors plot this for beta = 0.3? If the system is too noisy (or there is too much delay), then the system cannot really be used for experiments.

We think that there may have been a misunderstanding here and apologize for this. The VR system has inherent noise in it (such as the small errors with realtime 3D tracking (Stowers et al., 2017)), therefore, we did not add more synthetic noise on top of that. Please refer to our answer to your previous comment, #9: "Line 200*: What is beta in this experiment? How was it chosen?". We agree that an assessment of the level of noise inherent in the VR system (or, a validation of the system) is necessary. Please see Stowers et al., 2017 that we cite (a paper about this specific VR technique) as well as a preprint: Li, Liang, Mate Nagy, Guy Amichay, Wei Wang, Oliver Deussen, Daniela Rus, and Iain Couzin. "Reverse engineering the control law for schooling in zebrafish using virtual reality." (2023)—a link to the paper: <https://www.researchsquare.com/article/rs-2801869/v1>. There we ran experiments recreating pairwise interactions between two real fish where the interaction was only in the virtual world (both fish were in separate VR systems). This serves as a proof of concept that although there is noise present in the VR system, it still allows the fish to behave and interact naturally in such social contexts. Once

more, please refer to our previous replies for specific excerpts that clarify β throughout the manuscript.

20. Line 326: Substitute “we also evaluated our in-silico model, “in-virtualis”,” by: we ran simulations.

We believe there is a misunderstanding here and apologise for this. We coined a new term “in-virtualis” for these unique experiments that we utilised here, which we believe will be followed by more in the future. This is in contrast to simulations that are run on a computer. With “in-virtualis” we mean a case where we simulate the behavior of an agent, but it is rendered in VR so it can interact with real animals, hence we go from in-silico (computer based) to in-virtualis (in VR).

21. Line 331*: Can the authors display confidence intervals?

We performed randomizations (as also suggested by another referee; added to the SI) to give a sense of the variation in the data. Please see S14 here or in the Appendix.

22. Line 349 - 350: The statement “if, prior to the turn, they exhibited a specific temporal relationship akin to what we found in the open loop experiments” is very vague: can the authors be more explicit please.

We thank the referee for bringing to our attention this issue. We clarified this part in the manuscript. The sentence is now (lines 349-351):

“We found that individuals were considerably more responsive to the direction change of their virtual partner, and thus able to maintain close spatial proximity to them, if, prior to the turn, they exhibited the specific temporal relationship that we found in the open loop experiments”

23. Lines 381 – 383: I do not understand what these two categories are.

As per above, we thank the reviewer for helping us sharpen this explanation. We hope that the clarification provided above is sufficient.

24. Line 384: Can the authors include confidence intervals and explain how they performed the proportion test.

We have provided (previously) an SI figure showing all the data—all the cases that we analysed are fully shown (i.e. the dynamics over time), to get a sense for the variability in these data. We now point to this more clearly in the main text.

Regarding the proportion test, we clarified this in the manuscript by providing the equation that we used (lines 635-642):

$$z = \frac{P_A - P_B}{\sqrt{\frac{P_A(1-P_A)}{n_A} + \frac{P_B(1-P_B)}{n_B}}}$$

“Where P is the proportion of values for each specific case. We then determine if the result is significant (the p-value) with the cumulative distribution function (CDF) of the standard normal distribution, evaluated at z.”

25. Line 396: “which is a more stable configuration—both for reciprocity, and also in terms of visual sensitivity and detection” is an interesting observation; can the authors expand on this please? Referring to [52] is good, but if the authors mention it, it should be clear to the general reader why this is the case.

We are glad that the referee finds this interesting. We agree that it should be expanded and clarified. This has helped us make these statements more approachable for a wide range of readers. Please see below (lines 384-393):

“We finish by returning to our observational data of pairs of RF, and ask whether different temporal coupling regimes might be associated with certain spatial configurations, and whether such spatial configurations may also impact information flow/influence. We find that when their coupling is approximately out-of-phase, they tend to swim side by side (Fig. 4C and SI Appendix Fig. S17; projecting the 2D distributing in each panel on a circular axis, and then computing a circular Kuiper test, we obtain $p=0.001$). We note that it has been suggested that such side by side swimming may indeed be beneficial for social influence. Firstly, it facilitates reciprocity of information flow (any other configuration would be asymmetric in this respect), and furthermore, based on geometric principles, it has been shown that this configuration can allow individuals to optimize their detection of both speed and heading changes of a partner by utilizing perceived angular velocity and loom (approaching/receding) on the retina, respectively [59].”

26. Line 540: These methods “Co-moving frame of reference” should be expanded upon as they are currently insufficient to understand what was done in detail.

We thank the referee for pointing our attention to this. We agree that it was somewhat cryptic. In the methods section, we now refer the reader to the specific SI figure that shows more clearly how this was derived.

27. Line 545: Could the authors provide their simulation code?

We agree with the referee that these codes (as well as analysis codes) could have been provided and we have now done this.

28. Line 566 - 567: What is the point of this statement regarding Newton’s Law? Do the authors want to define what non-reciprocity is or what is an unbalanced system (in the sense of having a resultant force acting upon it)?

We agree with the referee that this part of the discussion might be a bit surprising. For the sake of developing the discussion broadly, we have chosen to also mention nonreciprocity as we think it is a new topic of interest in many-body systems and that our work touches upon it to some extent. Therefore, we wanted to define what reciprocity (or nonreciprocity) is in the physical sense, which is a resultant force acting upon a force, and how that differs from reciprocity in the behavioral sense (which was central to our study). Here is the updated relevant text (lines 400-402):

“Nonreciprocity, in the physical sense (the force exerted by one body on another wouldn’t be “reciprocated” equally) has recently been highlighted as a key ingredient in out-of-equilibrium systems [60]. Although behavioral reciprocity isn’t strictly equivalent, here we provide, previously unattainable, evidence of how a real system operates in this regard.”

29. Line 577 – 579: What about other fish swimming or vocalizations? Comparing juvenile zebrafish to fireflies (only) seems slightly random.

We agree with the referee and thank them for pointing this out. We have now added some clarification to connect this point better to the rest of the text. The text now reads (lines 412-449):

“This is in contrast with other animal collectives displaying “coupled oscillator” dynamics such as certain firefly swarms, where one-way interactions are sufficient—a firefly can entrain to a periodic artificial light [61]. One reason for this difference could be the fact that these fireflies are relatively isochronous, which can enable a nonreciprocal agent (not influenced by the firefly) and the firefly to fall in step.”

Reviewer #4 (Remarks to the Author):

Thank you for the opportunity to review the manuscript titled “Revealing the mechanism and function underlying pairwise temporal coupling in collective motion”. In this work, the authors address an intriguing, and to my knowledge, previously unexplored research question: How do animals in collective motion time their actions with respect to one another, and what functional implications does this temporal coupling bear? By employing an impressive combination of experiments, models, and data analytics, the authors present findings that will be of interest to a broad audience. Primarily, they discovered that schooling zebrafish larvae tend to alternate the timing of their tailbeats while swimming in pairs. They also introduce a straightforward rule explaining this synchronization, emphasizing that temporal coupling between two schooling fish emerges only when both adhere to this rule. The authors also demonstrate the functional significance of these behaviors, demonstrating that temporal coupling allows fish to avoid sensory limitations that would manifest if two fish were to accelerate simultaneously.

We thank the reviewer for their appreciation of our research, and due to their careful reading of the manuscript, they have helped us in improving it substantially.

This research is compelling and promises to spur further work on the implications of temporal coupling on collective behavior in diverse contexts and scales. Nonetheless, I have several reservations that require attention:

1. Concerning beta:

- The statement: “We find that, as in our virtual reality experiments with unidirectional information flow (Fig. 2D), that for relatively weak social coupling, and correspondingly relatively high intrinsic stochasticity (β), that unidirectional information flow is insufficient to allow A to achieve an out-of-phase coupling with B” is confusing because the authors set $\beta = 0.3$, which would seem to imply the opposite (they follow social cues more often than not).
- In the Supplementary Information, for example, Figure 7, the effect of changing beta is the opposite of what I would expect. $\beta = 1$ should correspond to completely random tailbeat timings, yet the bottom right panel shows the opposite. Perhaps the given definition of beta was flipped?

The reviewer is correct, there was an inconsistency in how the β parameter was defined. We thank the reviewer helping us by spotting this. We have now clearly defined β and consistently used that definition. Now $\beta = 0$ is completely random behavior, up to $\beta = 1$ which would mean responding according to the slope of the PRC.

Please refer to our reply to referee #1 comment 1 regarding our further analyses and explanations that clarify β throughout the manuscript.

2. The manuscript would benefit from justifying the selection of beta in the main text. If the necessity of two-way information flow for temporal coupling is contingent on the beta value, then it's crucial to use a beta value consistent with Panel 1E. Visually, the data seems better represented by strong social coupling. A potential approach could be estimating beta by minimizing the KL divergence of the modeled and observed bout duration distribution as a function of tau.

We are afraid that perhaps there was a misunderstanding here and apologize for this. A better and more thorough explanation of β is provided now in the paper (see here: referee #1 comment 1).

Panel 1E is unrelated to β —this is purely data analysis of real fish data (i.e. no model). β is introduced once we model the system with a computational model.

Overall, one could try to find the best fitting β value as suggested by the reviewer. We show that in a relatively large range of β values, similar behavior is reproduced. $\beta=0.3$ is just an example.

3. The main text should also incorporate a discussion on the implications of varying beta. How does the strength of social coupling affect temporal coupling in scenarios of both one-way and two-way information flow?

As mentioned in previous comments—a better and more thorough explanation of β is now provided in the revised manuscript and in our reply to referee #1 comment 1. This part was previously left mostly in the SI. But we have now discussed this (and the results of additional model analyses) in the main text.

Presentation clarity:

4. The interpretation of certain results could be made clearer. For instance, lines 173-174 state, “fish are found to exhibit prominent temporal coupling with respect to the timing of their bursts, exhibiting a characteristic (shared) time-lag, and out-of-phase (i.e., alternating) relationship, between bursts.” It would be clearer to reference Figure 1C here and further spell out how this is demonstrated by Figure 1C.

We thank the referee for this comment. We agree that this can ease the reading of this section. We added this explanation to the manuscript. Thus, it now reads (lines 169-173):

“...fish are found to exhibit prominent temporal coupling with respect to the timing of their bursts, exhibiting a characteristic (shared) time-lag, and out-of-phase (i.e., alternating) relationship, between bursts (as evident in the symmetric curve in Fig. 1C”

5. Furthermore, the section “One-way information flow...” would benefit from explicitly stating your expectations for Figure 2D if one-way information flow was sufficient for temporal coupling. For example, something like: “If one-way information flow is sufficient for temporal coupling, we would expect the timing of the focal fish’s tailbeats to alternate with those of the virtual fish. Thus, we would expect to see a drop in the correlation function at values of tau near zero, and peaks in the correlation function when tau is equal to half the beat frequency of the virtual fish. Instead, we find...”

Once again, we thank the referee also for this comment. We agree that this can ease the reading of this section. We added this explanation to the manuscript (lines 265-267):

“In other words, if one-way information flow is sufficient for temporal coupling, we would expect the timing of the focal fish’s tailbeats to alternate with those of the virtual fish. Thus, we would expect to see two peaks in the correlation function forming a symmetry about $\tau = 0$.”

6. What is the benefit to showing the correlation functions over negative beat timing differences?

That is a good question. By calculating a continuous correlation function, we can reveal patterns that go beyond one beat (and in both directions, not just negative), as evident in our one-way experiments. Furthermore, as we are dealing with non-isochronous signals, this gives the most accurate representation.

Intuitive explanation of mechanisms:

7. While the authors do an impressive job combining data analytics, models, and VR experiments to demonstrate that two-way information flow is needed for temporal coupling, I am still left wanting an intuitive explanation for why one-way information flow is insufficient for coupling. Given the authors have a simple model, it should be possible to delve deeper into the mechanisms underlying the results.

We thank the reviewer for this useful suggestion. We ran multiple different versions of the model, and present more analyses to further elucidate the dynamics of the system. Please see our responses to referee #1 comment 1.

8. Figure 3:

There's a noticeable difference between the y-axis scale in panel B compared to A and C. Is this merely due to varying bin sizes, or is there another factor?

Yes, we agree with this comment. We think that this indeed is mostly due to the different values/bin sizes on the x axis (2 order of magnitude difference).

9. For panel C, it isn't entirely convincing that it captures the essential features in panel A. It would be useful to have a null distribution to compare the line in C to, for example, by a randomization method.

To accompany this, we now performed randomizations (added to the SI) to give a sense of the variation in the data. Please see S14 below or in the Appendix.

10. It might also be worthwhile to discuss why the strength of the pattern appears much weaker in the VR experiment compared to the RF experiments.

We agree, and added more to the discussion regarding this. Please see below the following excerpt (lines 324-330):

"We note that while we observe a similar coupling dynamic as with real fish pairs (Fig. 3A), and in the model of bidirectional coupling (Fig. 3B), in the hybrid approach (real fish interacting with virtual fish exhibiting the rules of the model), the data are more noisy (Fig. 3C). This is due to the necessity of realtime tracking (which inevitably results in some tracking errors) and the fact that the motion of fish is not entirely determined by temporal dynamics (see the following section). Nonetheless, we still observe the main features (evident by the peaks in Fig. 3C) as found in the real system and model."

Minor comments:

11. Line 169: The assertion about the temporal structure in relation to Fig 1D needs validation.

We agree with the referee that there was room for improvement here, and have revised the sentence (lines 166-168):

"While each fish exhibits variable speed over time (due to their gait), the time intervals (lags) between successive bursts also exhibit a broad distribution of values (Fig. 1D)."

12. Figure 1E: A colorbar inclusion would greatly enhance the interpretation.

We agree and have added a colorbar, see the updated figure here or in the updated manuscript.

In summary, this study holds significant potential. With a few modifications, I believe it could serve as a cornerstone in understanding collective motion.

Thank you for your valuable insights.

FIGURES:

Fig. 1:

Fig. 2:

Fig. 3:

Fig. 4:

A VR experiments

Fig. S10:

Fig. S11:

Fig. S12:

Fig. S13:

Fig. S14:

REVIEWER COMMENTS

Reviewer #1 (Remarks to the Author):

I am satisfied with the authors' responses and their revision. In my opinion, the paper can now be published in Nature Communications.

Reviewer #2 (Remarks to the Author):

The authors have fully answered all my comments and questions. Moreover, they have extended the theoretical analysis and they have added interesting results on the stability and the transition.

As I had written in my first review: I strongly recommend this outstanding paper for publication.

Reviewer #4 (Remarks to the Author):

I think the authors did a good job addressing my concerns. However, I still have concerns about the authors' interpretation of Figure 3C. The authors now provide results from randomization simulations for Figure 3C, presented in Figure S14. However, for a better comparison, it seems appropriate that the randomization results in Figure S14 should be included within Panel 3C. This would allow readers to interpret the line in Figure 3C in the context of the underlying variability in the data. It appears that the line in Figure 3C falls well within the range of variability observed in the randomization simulations in Figure S14. Nonetheless, the authors conclude, '...we still observe the main features (evident by the peaks in Fig. 3C) as found in the real system and model.'

Perhaps I am misinterpreting the results, but to my eye, it seems that outcomes like the one presented in Fig 3C could also be generated due to randomness alone. Thus, I don't find Figure 3C to be particularly strong evidence supporting the authors' conclusion. This does not necessarily mean that the authors' conclusion about bidirectional interactions being required for temporal coupling is incorrect. It could be that lags, noise, and other limitations of the VR setup result in fish not responding to VR fish in a manner similar to real fish. However, the results in Fig 3C do not seem to

support the authors' hypothesis, considering how easily such a pattern can be generated by randomized data.

REVIEWER COMMENTS

Reviewer #4 (Remarks to the Author):

I think the authors did a good job addressing my concerns. However, I still have concerns about the authors' interpretation of Figure 3C. The authors now provide results from randomization simulations for Figure 3C, presented in Figure S14. However, for a better comparison, it seems appropriate that the randomization results in Figure S14 should be included within Panel 3C. This would allow readers to interpret the line in Figure 3C in the context of the underlying variability in the data. It appears that the line in Figure 3C falls well within the range of variability observed in the randomization simulations in Figure S14. Nonetheless, the authors conclude, '...we still observe the main features (evident by the peaks in Fig. 3C) as found in the real system and model.

Perhaps I am misinterpreting the results, but to my eye, it seems that outcomes like the one presented in Fig 3C could also be generated due to randomness alone. Thus, I don't find Figure 3C to be particularly strong evidence supporting the authors' conclusion. This does not necessarily mean that the authors' conclusion about bidirectional interactions being required for temporal coupling is incorrect. It could be that lags, noise, and other limitations of the VR setup result in fish not responding to VR fish in a manner similar to real fish. However, the results in Fig 3C do not seem to support the authors' hypothesis, considering how easily such a pattern can be generated by randomized data.

We thank the reviewer for their appreciation of our responses to their comments. We also agree with the new comment above, that the previous presented result in Figure 3C wasn't sufficiently strong evidence to support our claims. Indeed, it was possible (although not probable) that it could be attributed to noise. This is, as the reviewer intuited, due to the difficulty of conducting such an ambitious realtime feedback experiment in VR.

Because we think this is an important validation experiment, we decided to identify, and reduce, an important source of noise in this experiment. Specifically, we were able to improve the algorithm employed to detect the timing of the tailbeat-bursts of the real fish (minima-detection), to make it more accurate. Since doing so would result in cleaner data, which cannot simply be combined with data from the previous experiment, we decided to discard the old (more noisy) data and conduct this experiment again. These new data are presented in the new Figure 3C. In addition, following the reviewer's suggestion, we now include the randomization results in that same figure. As can be seen the peaks of the correlation function now extend well beyond the range of variability found in the randomizations. Please see the pasted text below, from the updated manuscript, as well as the updated the respective figure, Fig. 3:

"We found two significant peaks in this correlation function, akin to what appears in real fish pairs and the model (Fig. 3C; Kolmogorov-Smirnov test comparing to randomisation: $p < 0.001$). Thus, our model—despite its simplicity—captures the essential features employed in zebrafish temporal coordination. Reciprocal coupling is necessary, and sufficient, to observe the temporal dynamics exhibited by pairs of individuals."

We once again thank the referee and all other referees for their highly constructive comments, which helped us to substantially improve our paper.

FIGURES:

Fig. 3.:

Fig. 3 Comparing the temporal swimming patterns of pairs of real fish (**A**), in simulation (**B**) and in interactive virtual reality (**C**). The normalized correlation function C_{ij} of the beats (see Fig. 1D) as a function of the mean beat timing difference τ . (**A**) Two real fish swimming together in the same arena. (**B**) Simulation of two interactive agents using the model with $\beta = 0.3$ for the stochastic parameter produces similar temporal patterns as for 2 real fish. (**C**) Results for real fish swimming with an interactive virtual fish whose reciprocal behavior is controlled by the model. Due to the small but inherent noise (and delay) in the VR system, here β was set to 1 to avoid adding additional noise. The curve in blue shows the actual correlation function, whereas shuffled pairs generated by randomization (randomizing the entire dataset 100 times, resulting in 100 different curves; see the methods section for more details) are in yellow.

REVIEWERS' COMMENTS

Reviewer #4 (Remarks to the Author):

I appreciate the authors' effort to redo the VR experiment. Their new result is convincing, and I have no further concerns about the manuscript. Great work!